# PyHopper - A Plug-and-Play Hyperparameter Optimization Engine

**Mathias Lechner**
MIT
mlechner@mit.edu

**Ramin Hasani**
MIT
rhasani@mit.edu

**Philipp Neubauer**
DatenVorspung GmbH
philipp.neubauer@datenvorsprung.at

**Sophie Neubauer**
DatenVorspung GmbH
sophie.neubauer@datenvorsprung.at

**Daniela Rus**
MIT
rus@mit.edu

## Abstract

Hyperparameter tuning is a fundamental aspect of machine learning research. Setting up the infrastructure for systematic optimization of hyperparameters can take a significant amount of time. Here, we present PyHopper, an open-source black-box optimization platform designed to streamline the hyperparameter tuning workflow of machine learning research. PyHopper's goal is to integrate with existing code with minimal effort and run the optimization process with minimal necessary manual oversight. With simplicity as the primary theme, PyHopper is powered by a single robust Markov-chain Monte-Carlo optimization algorithm that scales to millions of dimensions. Compared to existing tuning packages, focusing on a single algorithm frees the user from having to decide between several algorithms and makes PyHopper easily customizable. PyHopper is publicly available under the Apache-2.0 license at https://github.com/PyHopper/PyHopper.

```python
import pyhopper                              ◄──────   pip3 install pyhopper

def objective(hparams):          ◄──────    Hyperparameters are dict objects
    model = build_model(hparams["size"],...)
    opt = Adam(hparams["lr"])

    train_loader, val_loader = ...
    # .... train model                               Use training code without changes

    val_accuracy = model.evaluate(val_loader)
    return val_accuracy
                                                     Pythonic search space definition
if __name__ == "__main__":
    search = pyhopper.Search(
        epochs = 100,                                Multidimensional array parameters
        size = pyhopper.int(100, 500),
        gain = pyhopper.float(0, 10, shape=(10,2)),
        opt  = pyhopper.choice("adam", "rmsprop"),
        lr   = pyhopper.float(1e-5, 1e-1, "0.1g"),   Limit search space via format strings
        ...                                          (e.g. "0.1g" → 1-significant digit
    )                                                              and loguniform)
    best_params = search.run(
        objective, "max",                            User-friendly way to set runtime
        runtime = "1h 30min",
        n_jobs="per-gpu"                             Run evaluations in parallel
    )                                                 on each available GPU
```

Figure 1: Visual abstract showing a typical use of PyHopper and some of its key features.

Has it Trained Yet? Workshop at the Conference on Neural Information Processing Systems (NeurIPS 2022).

# 1 Introduction

In this work, we introduce PyHopper, a hyperparameter optimization (HPO) platform tailored to the optimization scenarios we encounter in machine learning research (e.g., training neural networks). In particular, our HPO platform allows us to streamline the hyperparameter tuning procedures and scale to hundreds of tuning tasks with minimal effort. The key strengths of PyHopper are:

- An intuitive interface that integrates with existing machine learning code with minimal change,
- A highly customizable and robust optimization algorithm based on sequential Markov-chain Monte-Carlo sampling that scales to millions of hyperparameters (HPs),
- Numerous utilities to streamline common use cases, such as 1-line multi-GPU setup, checkpointing, pruning, and runtime scheduling.

Numerous hyperparameter optimization algorithms have been proposed in the literature, each specialized for specific use cases and applications. Moreover, many publicly available HPO packages implement these algorithms. In this section, we first discuss the most important HPO algorithms and how they compare with each other. In the second part, we describe common HPO packages for Python and highlight their differences from PyHopper.

## 1.1 Hyperparameter Optimization Algorithms

**Grid Search** is arguably the most basic HPO. As its name suggests, Grid search spans a grid over the parameter space and evaluates every intersection point of the grid. The main advantage of Grid search is that it explores all parts of the configuration space. Thus it does not get easily trapped in local optima. However, the major bottleneck of Grid search is that its complexity scales exponentially with the dimension of the configuration space, thus beeing suitable for low dimensional configuration spaces, i.e., typically 2 or 3 hyperparameters.

**Sequential Model-Based Optimization** (SMBO) [7] is a powerful black-box optimization paradigm. The key idea of SMBO is to fit a simple-form surrogate model to the already evaluated points to interpolate between unexplored parts of the configuration space. The easily computable optimum points of the surrogate model are evaluated on the actual objective function. **Bayesian Optimization** (BO) extends SMBO by fitting distributions instead of deterministic functions to the evaluated points of the configuration space. The key benefit of BO over SMBO is that it allows modeling the uncertainty about the interpolated parts of the configuration space.

**Tree-structured Parzen Estimator** (TPE) [2] is a sequential model-based optimization algorithm that can handle conditional configuration spaces efficiently. An example of such a conditional configuration would be the number of layers and corresponding hidden units in each layer of a neural network. Particularly, the number of layers in the fifth layer is only needed if the number of layers exceeds 4.

**Random Search** (RS) is another straightforward black-box optimization baseline. RS samples candidate solutions from a uniform distribution over the entire configuration space. Despite its simplicity, RS can be competitive and outperform alternative algorithms in high-dimensional configuration spaces [3].

**Markov-chain Monte-Carlo** (MCMC) is a family of methods for sampling from a probability distribution that cannot be described in a simple form explicitly but as the stationary distribution of a Markov-chain. The most fundamental MCMC optimization method takes the current best configuration and generates a new sample by adding random noise to it. Such procedures are also referred to as *local search* or *hill climbing* in the optimization literature. **Simulated Annealing** (SA) [9] further extends this idea by continuously decreasing the variance of the added noise, i.e., the "temperature", and also keeping worse new samples with a certain probability.

## 1.2 Hyperparameter Optimization Packages

There exist a large set of publicly available hyperparameter tuning packages. Here, we briefly discuss some of the most common tools and highlight some of their unique features.

**HyperOpt** [5] is a hyperparameter tuning framework that provides an implementation of the Random Search and the Tree of Parzen Estimators optimization algorithms. The specialty of HyperOpt is that parallelization is supported via Apache Spark or in a custom way through a database. This allows HyperOpt to integrate with an Apache Spark cluster easily at the additional cost of effort to set up and maintain the Apache Spark cluster.

**Optuna** [1] is a hyperparameter tuning framework developed by Preferred Networks. Optuna implements many common optimization algorithms and supports parallel evaluation through a MySQL database to which remote evaluation workers can connect. The main focus of Optuna is on experiment tracking and visualization of evaluated configurations.

**NeverGrad** [16] is a black-box optimization library developed by Meta. It implements various gradient-free optimization algorithms and allows executing multiple configurations in parallel.

**keras-tuner** [15] is a hyperparameter tuning library building on top of the Keras API and Tensorflow 2. The package implements common tuning algorithms, including Random Search, Bayesian optimization, and the HyperBand algorithm [12]. The major limitation of Keras-tuner is that it does not support running multiple evaluations in parallel.

**Autotune** [10] is an HPO platform focused on large-scale tuning of traditional models. Autotune implements several evolutionary sampling algorithms that can be combined during the search.

**Dragonfly** [8] is a black-box optimization package that implements variants of Bayesian Optimization algorithms. On-machine multiprocessing for parallel evaluation is available in Dragonfly.

**Ray Tune** [13] is the hyperparameter library building on top of the Ray distributed computing framework. Ray Tune provides an enormous set of different hyperparameter tuning algorithms. Moreover, Ray Tune can serve as a distributed evaluation engine for other hyperparameter tuning tools such as Optuna, Dragonfly, and HyperOpt. While Ray Tune is relatively flexible in terms of possible parallelization and tuning procedures, the large number of available algorithms can overwhelm the user by creating the meta-problem of finding the best HPO algorithm.

## 2 PyHopper's Pythonic API Design Characteristics

We design PyHopper's API to have a flat learning curve and allow integration with existing code with minimal effort. Our approach to achieving this goal is to make use of concepts that most Python developers are already familiar with, so working with PyHopper feels natural.

**PyHopper's optimization algorithm.** PyHopper's optimization algorithm is a 2-phase MCMC sampler. In phase 1, a random search samples candidates uniformly to get a coarse view of the objective landscape. In phase 2, the search area is continuously narrowed down to the neighborhood of the current best candidate. A pseudocode description of PyHopper's optimization algorithm and further details can be found in Appendix A. The advantage of having a single algorithm is that the user does not need to bother selecting a suitable algorithm, i.e., replacing the problem of finding the best hyperparameters with the meta-problem of finding the best HPO algorithm. Moreover, custom datatypes and sampling strategies are straightforward to implement in PyHopper.

**Separating training code from HPO code** Typically, the training pipeline is developed before any thoughts are given about the hyperparameter optimization process. Thus, for a streamlined HPO interface, the training code should not be aware of any HPO. PyHopper realizes this by storing hyperparameter candidates as `dict` objects.

**\*\*kwargs configuration space and reusing familiar Python keywords.** PyHopper's configuration space, i.e., the datatypes and value ranges of the HPs to consider, is defined by passing named arguments to the initialization of the search object. This results in easily understandable and maintainable code. Most Python developers are familiar with the built-in type-casting functions `int()` and `float()`, which is why we use them for defining the search space. Additional properties, such as quantization and uniform vs log-uniform distributions, can be enabled via optional arguments.

**Log-uniform and quantized distributions via format string.** Most Python developers are somewhat proficient with Python format strings of floating-point numbers. For instance the `":0.2f"` formats a floating-point number to two decimal digits after the comma, or `":0.1g"` to one significant digit in scientific notation (e.g., 3e-5). We re-purpose this concept to allow quantization and logarithmic sam-

Table 1: Results of our experimental comparison of common hyperparameter tuning packages. The objective function was set to maximize the validation accuracy and minimize the validation mean-squared error (MSE) for the IMDB and Walker2D task respectively. For the tools supporting 1-line parallel multi-GPU execution (ray-tune and PyHopper), parallel execution was enabled. Optuna [1], ray-tune [13], and PyHopper demonstrate competitive optimization performance.

| Dataset | IMDB [14] | | Walker2D [11] | |
|---|---|---|---|---|
| HPO platform | Best validation accuracy | Runtime (minute) | Best validation MSE | Runtime (minutes) |
| Optuna [1] | 87.47% | 823 | **0.863** | 42 |
| HyperOpt [4] | 50.00 % | 808 | 1.956 | 42 |
| ray-tune [13] | 87.46% | 104 | 0.897 | 23 |
| PyHopper (ours) | **87.98**% | **80** | 0.878 | **20** |

pling of the parameter space. For example, passing `":0.2f"` to a float-type PyHopper hyperparameter quantizes it to two decimal digits and uses a (linear) uniform search space. Contrarily, `":0.1g"` makes the float parameter sample from a log-uniform distribution quantized to one significant digit.

**Additional features of PyHopper.** PyHopper provides a large set of built-in utilities designed for making the HPO process as smooth as possible. For instance, on multi-GPU systems, Py-Hopper can evaluate candidates in parallel processes, each using a single GPU (via setting the `CUDA_VISIBLE_DEVICES` environment variable of the parallel subprocesses). Additionally, PyHopper runtime argument accepts strings such as `"24h"` or `"1day 2h 30min"`, which allows directly forwarding arguments from the command line (via `argparse` instance). Moreover, PyHopper implements a pruning algorithm similar to those found in Optuna [1], which allows stopping unpromising candidates early in their evaluation, e.g., if the validation accuracy after the first 10 epochs is not within the top quantile. Further features and details about PyHopper can be found in Appendix C.

## 3 Experiments

We set up an experimental evaluation to benchmark four popular hyperparameter optimization platforms: Optuna [1], HyperOpt[4], ray-tune[13], and PyHopper. For a fair comparison, we define the exact same configuration spaces for all methods and allow each method to sample 30 hyperparameter configurations in total. The objective function that should be optimized by the tools consists of training a Transformer model [17] on the IMDB sentiment analysis dataset [14]. Our second experiment concerns the training of an LSTM network on the Walker2D kinematics modelling dataset [11]. The hyperparameters include, among others, the learning rate, number of attention heads, size of the LSTM cell [6], and dropout rate applied to the word embedding. Random seed was fixed for the training (weight initialization and dataset shuffling). The full configuration space can be found in Table 3 and Table 4 in Appendix F.

Both ray-tune and PyHopper provide 1-line multi-GPU parallelization capabilities, which we enable for the evaluation. This significantly reduces the runtime on our machines (8 Titan RTX GPUs for the IMDB task, and 2 A6000 GPUs for the Walker2D task), however, it potentially limits the optimization as strictly sequential testing of hyperparameters results in the maximum information being available for generating informed new candidates.

The results in Table 1 show that Optuna, ray-tune, and PyHopper could find competitive performing hyperparameter settings for both tasks. Moreover, the runtimes indicate that PyHopper concluded its search process the fastest. In particular, for IMDB, PyHopper's runtime is more than 10x faster than Optuna and HyperOpt, and 1.2x better than ray-tune. Moreover, we observed that HyperOpt was not able to find good hyperparameter candidates. We hypothesize that HyperOpt's Bayesian optimization engine did not allocate the 30 available samples to properly cover the search space.

## 4 Conclusions

PyHopper is a customizable, open-source, and plug-and-play hyperparameter optimization engine, that can be integrated with advanced training jobs with minimal effort and low cost, generating competitive models compared to existing well-established packages.

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

## A   Optimization algorithm

PyHopper revolves around a single optimization algorithm based on MCMC sampling. The algorithm consists of two sequential phases, i.e., an exploration and an exploitation phase. In phase 1, a Random Search draws random samples uniformly over the entire configuration space. The main idea of phase 1 is to gather information about the objective surface, i.e., which parts of the configuration space seem promising and which do not.

In the second phase, a sequential MCMC sampler takes the current best configuration and generates new samples by adding random noise. The principle insight of phase 2 is to make incremental but consistent improvements to the current best solution. To illustrate an example, phase 1 is about filtering out what hyperparameter combinations do not work at all, while phase 1 is about the details of the optimal ones, e.g., whether the best learning rate is 0.01 or 0.02. A pseudocode representation of PyHopper's algorithm is shown in Algorithm 1.

**Algorithm 1** High-level description of PyHopper's MCMC sampling algorithm (maximization)

---

**Input** Parameter space $\Omega$, objective function $f : \Omega \to \mathbb{R}$
$\theta_1, \ldots \theta_k \leftarrow$ random samples from $\Omega$          $\triangleright$ Random search (phase 1)
$\theta_{best} \leftarrow argmax_{\theta_i}\{f(\theta_i\}$
temperate $\tau \leftarrow 1$
**while** not timeout **do**
    $\theta \leftarrow \theta_{best}$ + random noise with temperature $\tau$        $\triangleright$ Local search (phase 2)
    **if** $f(\theta) > f(\theta_{best})$ **then**
        $\theta_{best} \leftarrow \theta$
    **end if**
    decrease temperature $\tau$
**end while**
**return** $\theta_{best}$

---

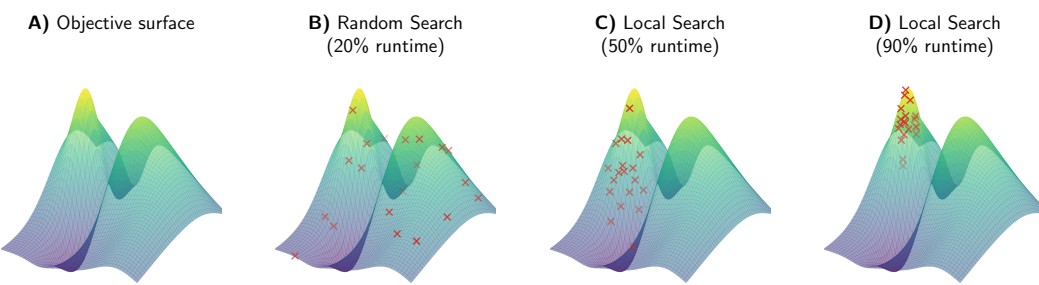

**A)** Objective surface    **B)** Random Search (20% runtime)    **C)** Local Search (50% runtime)    **D)** Local Search (90% runtime)

Figure 3: Example illustration of a 2-dimensional optimization problem and how PyHopper's optimization algorithm gradually narrows down the search area. **A)** Objective surface. **B)** Evaluated points during the Random Search (phase 1). **C)** Evaluated points during the beginning of the Local Search (phase 2). **D)** Evaluated points during the end of the Local Search (phase 2).

PyHopper adopts the idea of simulated annealing and gradually decreases the magnitude of the noise over the runtime of the tuning process. Phase 2 is referred to as local sampling due to narrowing down the search area locally around the current best solution. The schematic of the two phases and corresponding scheduling is shown in Figure 2. By default, Py-Hopper spends 25% of the runtime performing random search (phase 1). The "temperature", i.e., the

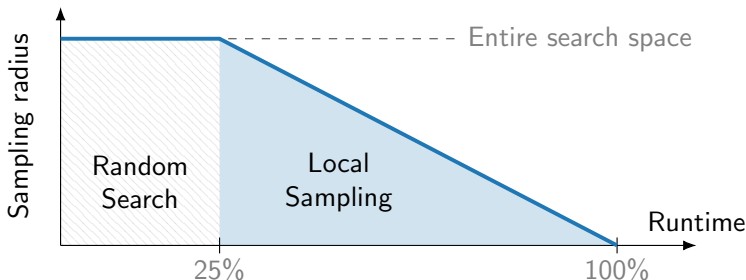

Figure 2: Visualization of PyHopper's scheduling. At the beginning, a Random Search uniformly samples candidates from the entire configuration space. During the second phase, the Local sampling procedure gradually narrows the search area around the current best configuration. By default, the Random Search is scheduled to account for 25% of the total runtime, while annealing of the Local sampling phase is scheduled for the remaining 75%.

noise variance, then linearly decreases over the remaining runtime in the second phase running the local search.

PyHopper requires the user to specify the target runtime of the tuning process, which provides two major benefits. First, it allows exact scheduling between the two phases and the annealing process of phase 2. This ensures that the tuning process spends sufficient time exploring the configuration space and exploiting the promising areas. The provided runtime comes with the additional benefit that we can schedule the tuning process to maximize the usage of available hardware. For instance, we can

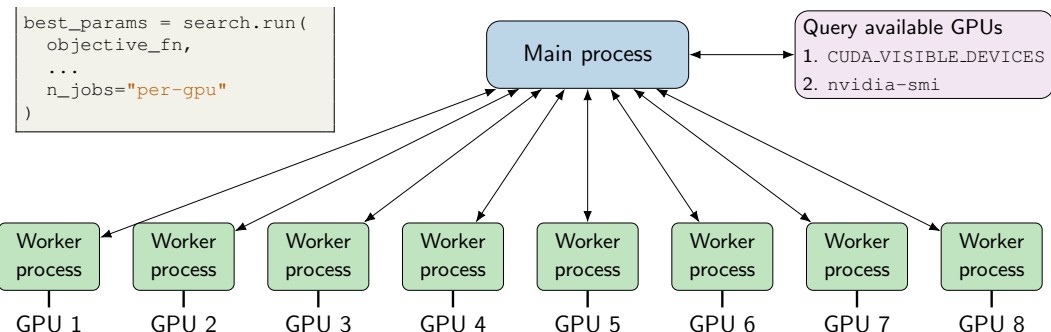

Figure 4: High level description the parallization built-in by PyHopper. If the `n_jobs` argument is set to `per-gpu`, the objective function is executed in parallel in subprocesses with only a single GPU visible to each process.

specify PyHopper to run overnight or over the weekend and be done by Monday morning. Figure 3 visualizes on an example how searchable area gradually focuses over the scheduled runtime.

The focus of PyHopper on a single optimization algorithm avoids the dilemma of having to decide between multiple tuning algorithms, i.e., the meta-problem of finding the best HP tuning algorithm, as well as it allows for streamlining the user interface.

PyHopper's algorithm is flexible and customizable. For example, we can skip phase 1 and directly let the local sampling algorithm improve on a set of hyperparameters the user provides. Such scenarios often occur when the user finds some decently working hyperparameter through a manual search. Moreover, PyHopper allows integrating custom sampling and local perturbation (i.e., mutation) strategies for special types of problems. For instance, the Travelling salesman problem (TSP) is an NP-complete combinatorial optimization problem that concerns finding the shortest roundtrip over a set of cities. With PyHopper, we can implement algorithms for heuristically solving the TSP with very minimal code.

# B  Parallelization

PyHopper can parallelize the evaluations of hyperparameter candidates by executing the objective function on multiple processes simultaneously. The sampling of new candidate solutions, scheduling, and invoking callback functions are all done in the main process. Thus, no code change is required by the user to use PyHopper's parallel executing engine.

PyHopper is primarily intended to run on a machine with multiple NVIDIA GPUs installed (single-machine multi-GPU). We can spawn a parallel evaluation process for each available GPU with a single argument. PyHopper takes care of detecting the number of installed GPUs and setting the environment variables accordingly. Multi node parallization may be included in future versions of PyHopper.

# C  API design

Machine learning research can involve an enormous amount of hyperparameter tuning. PyHopper's API is designed to minimize the necessary changes that have to be made to the training pipeline and to simplify the integration of the tuned hyperparameter to other code. Particularly, the user interface of PyHopper aims to remove the friction between the ML code and the hyperparameter tuning code.

## C.1  Separation of concerns

Our key idea is that any ML code, i.e., for both training and usage afterward, should be able to run without any dependency on the hyperparameter tuning package. In particular, this means that the objective function, i.e., the training and validation, should contain any call to a function from the tuning package. Consequently, PyHopper does not implement a *define-by-run* API as done in other

**A)**

```python
hparams = {
    "lr" : 0.001,
    "size": 512,
    "weight_decay": 1e-6,
    ...
}

model = build_model(hparams["size"],...)
opt = Adam(hparams["lr"])

train_loader, val_loader = # ....
# .... fit model

val_accuracy = model.evaluate(val_loader)
# our objective to maximize
```

**B)**

```python
def objective(hparams):
    model = build_model(hparams["size"],...)
    opt = Adam(hparams["lr"])

    train_loader, val_loader = # ....
    # .... fit model

    val_accuracy = model.evaluate(val_loader)
    return val_accuracy

if __name__ == "__main__":
    search = pyhopper.Search(
        size  = pyhopper.int(100,500),
        lr = pyhopper.float(0.0001,0.1),
        ...
    )
    best_params = search.run(
        objective, "max",
        runtime = "1h 30min",
        n_jobs="per-gpu"
    )
```

Figure 5: Example code snippet of how PyHopper integrates with existing ML code. **A)** Typical machine learning code for training a neural network. **B)** Adapted code for hyperparamter tuning with minimal required changes.

Table 2: List of supported datatypes in PyHopper and corresponding examples.

| Definition | Samples |
|---|---|
| Integer parameters (uniform) | |
| PyHopper.int(100,500) | 350, 250, 500, ... |
| PyHopper.int(100,500, multiple_of=100) | 400, 200, 100, ... |
| PyHopper.int(0,10, shape=3) | (5,2,7), (10,2,6), ... |
| Integer parameters (logarithmic) | |
| PyHopper.int(2,64, power_of=2) | 8, 4, 32, ... |
| Float parameters (uniform) | |
| PyHopper.int(0,1) | 0.5434, 0.83934, ... |
| PyHopper.int(0,1, precision=1) | 0.5, 1.0, 0.3, ... |
| PyHopper.int(0,1, "0.1f") | 0.5, 1.0, 0.3, ... |
| PyHopper.int(-10,10, shape=2) | (-6.22343, 1.5234), (0.3632,-8.90331), ... |
| Float parameters (logarithmic) | |
| PyHopper.int(1e-5,1e-3,log=True) | 0.00023784, 0.000072342 ... |
| PyHopper.int(1e-5,1e-3,log=True,precision=1) | 2e-4, 5e-5, 8e-4, ... |
| PyHopper.int(1e-5,1e-3,"0.1g") | 2e-4, 5e-5, 8e-4, ... |
| Set parameters (unordered) | |
| PyHopper.choice(["adam","sgd","rmsprop"]) | "sgd", "adam", ... |
| Set parameters (ordered) | |
| PyHopper.choice([1,10,100],is_ordinal=True) | 10, 1, 100 , ... |

```python
import pyhopper
import argparse

def objective(hparams):
    # .... build and fit model
    return val_accuracy

if __name__ == "__main__":
    parser = argparse.ArgumentParser()
    parser.add_argument("--runtime", default="2h")
    parser.add_argument("--n_jobs", default="1")
    args = parser.parse_args()

    search = pyhopper.Search(
        ...
    )
    best_params = search.run(
        objective, "max",
        runtime = args.runtime,
        n_jobs=args.n_jobs
    )
```

```
$ python3 script.py --runtime 12h   --n_jobs per-gpu
```

Figure 6: Example of how PyHopper's run parameters are supposed to be forwarded from the command line arguments.

hyperparamter tuning packages [1, 15]. Instead, the interface between PyHopper and ML pipeline is represented by a ython dictionary (i.e., dict) object containing the hyperparameters. Additionally, this design choice provides the advantage that the tuned hyperparameters in the form of a Python dictionary can be easily stored, logged, and examined by the user.

The required adaptions of typical ML training pipelines to PyHopper are to wrap the training and validation into an objective function, define the configuration space, and run the search. The search space is set up by defining a template hyperparameter dictionary. An example is shown in Figure 5.

## C.2  Helper functions

PyHopper implements several helper and utility functions that simplify common tasks involved in hyperparameter tuning. For instance, instead of defining the runtime of the tuning process in seconds, a string can be provided that will be parsed, supporting multiple units of time, e.g., hours or minutes. Another example is the njobs argument, which, if set to "per-gpu" will take care of querying how many GPU devices are available on the system and scale the tuning process to run an evaluation on each device in parallel.

## C.3  Customization

PyHopper allows defining custom parameter types by defining custom sampling and mutation strategies. Additionally, PyHopper allows pruning candidates during the evaluation, e.g., if the validation accuracy does not reach a certain threshold in the first few epochs. Live feedback of evaluated candidates and current best configurations can be streamed through callback functionalities in PyHopper. For instance, PyHopper provides a built-in checkpointing mechanism that continuously saves evaluated candidates and corresponding objective values in a file so that no information is lost if the machine crashes.

**A)**

```
import pyhopper

search = pyhopper.Search(
  lr_lin = pyhopper.float(1e-5, 1e-1),
  lr_log = pyhopper.float(1e-5, 1e-1, log=True)
)
# log=False is default
```

**B)**

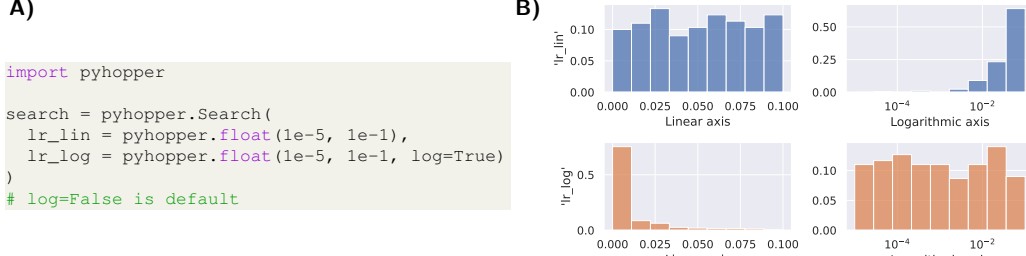

Figure 7: Difference between a linear and logarithmically distributed float parameter. **A)** Logarithmical sampling of a float parameter can be enabled via the `log` argument. **B)** Resulting histogram of sampled values. The top row shows a linear and the bottom row a logarithmically distributed parameter. The two plots on the left use a linear x-axis, while the two plots on the right use a logarithmic x-axis.

### C.4  Pruning algorithms

The training of neural networks is an inherently stochastic process. For it instance, the random seed for the weight initialization affects the final accuracy of a model. This can be problematic for hyperparameter tuning, as we cannot fully trust the objective function always returns the same value. In particular, some evaluations might be lucky and report a bit higher performance in the objective function due to a particularly good random seed.

Fixing the random seed in the training process avoids the stochasticity of the objective function. However, a fixed seed makes the tuning process potentially overfits the hyperparameters to the specific train-validation split. As a result, transferring the hyperparameters to a different train-validation split may result in a drop in performance.

A more reliable way to deal with this problem is to evaluate every HP configuration several times and report the mean. However, this significantly increases the computational cost of the tuning procedure. To counteract this cost explosion while maintaining the reliability of testing a configuration with several random seeds, we can employ pruning algorithms that stop the evaluation process of unpromising candidates already after their first evaluation. In particular, if the first evaluation of an HP candidate indicates that this configuration has very little chance of being the best hyperparameter, the pruning algorithm will stop the remaining evaluations and discard the HP candidate.

The API design for the pruning interface was inspired by Optuna [1].

## D  Examples

In this section we provide the most fundamental examples demonstrating the real-world use of PyHopper.

### D.1  Available datatypes

In Table 2, we list the available datatypes of hyperparameters in PyHopper. Moreover, Table 2 demonstrates the most common customization patterns and corresponding samples.

For an efficient search, we distinguish between linearly and logarithmically distributed hyperparameters. The default case is parameters with a linear search space and uniform density. This can be problematic for parameter spaces that range over several orders of magnitude. For instance, for a learning rate parameter spanning between $10^{-5}$ and $10^{-1}$, there is a 50% chance that a uniform sample will be greater than $5 \cdot 10^{-2}$, i.e., approximately the center of the interval $[10^{-5}, 10^{-1}]$. Consequently, the optimization algorithm is biased toward sampling large values and might not explore values are the lower end of the spectrum with the same frequency. A logarithmic sampling distribution might resolve this issue and is therefore preferred for parameters that span multiple orders of magnitude. In Figure 7, we visualize sampling differences between a linear (default) and logarithmically distributed parameter.

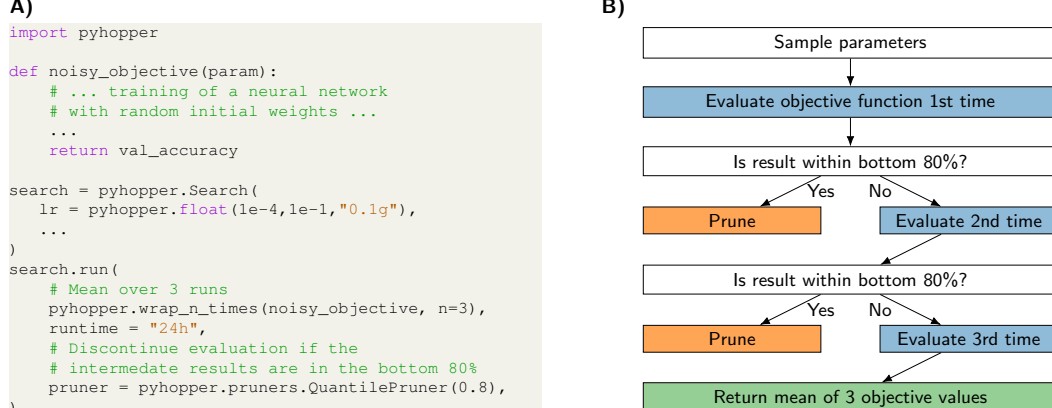

Figure 8: Example of how to deal with stochastic objective function and instantiate a pruning algorithm in PyHopper. **A)** Code snippet showing that only two lines of code are required to evaluate the mean of a noisy objective function and instantiate a pruning algorithm that prunes the candidate if the intermediate results are not within the top 20%. **B)** Corresponding flow-chart involved in the pruning decision.

### D.2 Command line arguments

PyHopper accepts its runtime and njobs argument in the form of a string that will be parsed. This allows directly forwarding command line arguments to PyHopper's run method and building an intuitive interface for the user with minimal code without limiting the freedom and flexibility of the developer. An example of this command line argument forwarding is shown in Figure 6.

### D.3 Noisy objective and pruning

As highlighted before, one way to deal with a stochastic objective function is to evaluate it several times and optimize the mean. Moreover, we mentioned pruning algorithms to discontinue candidates that turn out to be unpromising already after the first evaluation. In Figure 8, we demonstrate that only two lines of code are required to implement this behavior. In particular, the PyHopper.wrap_n_times helper function wraps the objective function into a loop evaluating a given number of times. Moreover, the intermediate results are sent to the pruning algorithm, which decides whether to continue or prune the candidate.

### D.4 Fault tolerance and preemptive compute instances

A typical hyperparameter tuning process can run for several days or even weeks. It is, therefore, necessary to prepare for unexpected events, such as power cuts or software bugs, to avoid a loss in data. Additionally, to optimize costs, we could run the hyperparameter tuning process on preemptive cloud machines (i.e., spot instances), which can be shut down anytime. PyHopper provides a checkpointing mechanism that continuously saves its internal state on the disk. Consequently, in case the tuning process is interrupted, e.g., by a power cut or spot instance shutdown, we can resume the tuning process from the last checkpoint. In Figure 9, we demonstrate how to use the checkpointing mechanism of PyHopper.

## E Limitations

There cannot be a perfect hyperparameter tuning package, as some features of what makes a good HP tuner might be contradictory. For instance, implementing several different optimization algorithms might be both an advantage and a disadvantage. Instead, each hyperparameter tuning package comes with tradeoffs that were made for specific application areas in mind. The main tradeoff for PyHopper is the focus on a single optimization algorithm. As highlighted before, having a single algorithm provides the advantage the user does not need to bother with which algorithm to

```
import pyhopper

search = pyhopper.Search(
    ...
)
opt_params = search.run(
    ...
    checkpoint_path = "ckpt_file.ckpt"
)
```

(a) In case a filename is provided, PyHopper will resume the search from the checkpoint if the file exists. Continuous progress of the search will be stored in the file.

```
import pyhopper

search = pyhopper.Search(
    ...
)
opt_params = search.run(
    ...
    checkpoint_path = "ckpt_dir/"
)
```

(b) In case a directory is provided, PyHopper will create a new checkpoint within the directory. Progress is stored in the new checkpoint file.

```
import pyhopper

search = pyhopper.Search( ...
)
search.load("ckpt_file.ckpt")
opt_params = search.run(
    ...
)
search.save("new_ckpt.ckpt")
```

(c) Manual checkpointing. Through the load and save functions manual checkpoints can be created and loaded.

Figure 9: Demonstration of different way to use PyHoppers checkpointing mechanism

Table 3: Configuration space of our experiment setup training a Transformer model [17] on the common IMDB sentiment dataset [14]

| Hyperparameter | Range |
|---|---|
| Learning rate | 1e-4 to 1-e2 (loguniform) |
| Learning rate deacy | 0.2 to 1.0 (quantized in 0.1 steps) |
| Warumup gradient steps | 100 to 1000 (quantized in 100 steps) |
| Learning rate decay every n-th epoch | 5,10 |
| Number of attention heads | 4 to 8 |
| Dimension per attention head | 16 to 128 (quantized in 16 steps) |
| Feedforward dimension | 64 to 512 (quantized in 64 steps) |
| Weight decay | 1e-6 to 1e-4 (loguniform) |
| Dropout rate | 0 to 0.3 (quantized in 0.1 steps) |
| Number of layers | 2 to 6 |
| Apply LayerNorm on word embedding | True, False |
| Word embedding dropout rate | 0 to 0.3 (quantized in 0.1 steps) |

run, i.e., the meta-problem finding the best hyperparameter tuning algorithms that finds the best hyperparameters. Moreover, the focus on the single MCMC sampler allows the user to easily customize the algorithm as there are fewer layers of abstraction than packages that implement several optimization routines. Nonetheless, PyHopper's optimization algorithm comes with a set of disadvantages. First, Bayesian optimization-based approaches might be preferable for lower dimensional problems and for problems where many candidate configurations can be evaluated. For such problem instances, enough information about the objective surfaces is available for a model-based algorithm to accurately approximate the black-box objective function via a surrogate model and overcome the problem of getting stuck in local optima. Moreover, running multiple hyperparameter algorithms in parallel might be feasible for problems where our compute budget is large enough. Consequently, packages that can try several different algorithms might have an advantage over PyHopper in such cases.

## F  Experiment details

The configuration space considered for both experiments are listed in Table 3 (IMDB task) and Table 4 (Walker2D task) respectively.

Table 4: Configuration space of our experiment setup training an LSTM network [6] on the Walker2D kinematics modelling dataset [11]

| Hyperparameter | Range |
| --- | --- |
| LSTM cell size | 64 to 512 (quantized in 64 steps) |
| Learning rate | 1e-4 to 1-e2 (loguniform) |
| Learning rate deacy | 0.2 to 1.0 (quantized in 0.1 steps) |
| Warumup gradient steps | 100 to 500 (quantized in 100 steps) |
| Learning rate decay every n-th epoch | 5,10,20,50 |
| Weight decay | 1e-6 to 1e-4 (loguniform) |

