# OpenReview forum: "PyHopper - A Plug-and-Play Hyperparameter Optimization Engine"
_NeurIPS.cc/2022/Workshop/HITY — HITY Workshop NeurIPS 2022_

### Official Review · Reviewer_AVqX · 2022-10-06
**An easy-to-use pythonic HPO library**

**Rating:** 1
**Confidence:** 4

**Review:**

This work presents PyHopper, a tool for automatic hyperparameter optimization in ML tasks that focuses on a pythonic interface to achieve a flat learning curve, and avoids introducing novel hyperparameters. It relies on a single two-phase MCMC algorithm that first explores the search space, before locally narrowing down the solution. Experiments suggest that PyHopper is competitive with, sometimes superior to, other HPO libraries, while simple to use.

---

Detailed comments:

The code snippet on page 1 demonstrates that PyHopper is easy to use. Maybe the authors could explain if it is possible to get access to the trained model of the best parameters, or if PyHopper simply returns the best hyperparameters, for which another round of training needs to be performed.

The paper provides a comprehensive description of existing HPO methods and libraries, and motivates PyHopper's design choices. The code in Fig. 1 suggests that PyHopper integrates with arbitrary ML frameworks, which could be highlighted more clearly.

I was wondering how PyHopper manages to distribute runs over devices without being agnostic to the chosen ML library. Maybe this could be briefly explained in the "Additional features" section.

The conclusion is quite generic and could be more specific about the unique features and performance aspects of the library.

---

Miscellaneous comments:

- Figure 1: Use consistent spacing convention in code, e.g. space between comma and next argument of a function.
- L18: integrates *with* existing...?
- Improve consistency of "Markov chain" vs. "Markov-chain"
- Improve consistency of "Monte-Carlo" vs. "Monte Carlo"
- L40: "they allow" → "it allows"
- L74: Add a reference for the HyperBand algorithm as it is not mentioned before.
- L110: "digits" → "digit"
- L121: "after the first epoch 10" → "after the first 10 epochs"
- L308: "Helpler" → "Helper"
- Table 1: Fix library names in caption

---

### Official Review · Reviewer_UerB · 2022-10-08
**A new and easily integrable hyperparameter optimization package**

**Rating:** 1
**Confidence:** 3

**Review:**

A new hyperparameter optimization package is proposed, which can be easily integrated into the training pipeline and with comparable performance to existing packages, while running fast. The results look promising and the content is presented clearly.

---

### Decision · Program_Chairs · 2022-10-20

Accept